# How to measure earnings surprises: Based on revised market reaction

Qin Pan [ID]*, Kai Huang

Institute of Chinese Financial Studies & School of Finance, Southwestern University of Finance and Economics, Chengdu, China

* 119020204046@smail.swufe.edu.cn

## Abstract

We investigate the robustness of earnings surprise measures in the context of a revised market reaction. While existing literature suggests that financial anomalies may distort cumulative abnormal returns (CAR) during annual announcements, our research proves that a revised market reaction offers a more accurate reflection of investor reactions to earnings correction. Specifically, we introduce an innovative adjustment to CAR using stock price jumps, and prove that the fraction of misses on the same side (FOM) provides a superior measure of earnings surprises. Furthermore, we find that investor trading patterns align with FOM, and the post-earnings announcement drift (PEAD) strategy based on FOM outperforms that based on analysts' forecast error.

**Data Availability Statement:** All relevant data are within the paper and its Supporting information files.

**Funding:** The author(s) received no specific funding for this work.

## 1. Introduction

A substantial body of literature shows that analysts' tendency to exhibit an optimistic bias in their earnings forecasts, revealing that the forecasts released by analysts are typically higher than the actual earnings [1–3]. The overly optimistic forecasts not only suggest that analysts place excessive weight on their private information [4], but also withheld negative information causing lower pricing efficiency in the stock market [5]. Nonetheless, analysts' earnings forecasts remain an integral benchmark for current scholars to measure the market consensus [6]. Much of the literature takes the average (the median or the latest) of analysts' earnings forecasts before the earnings announcement as the market consensus, and then calculates earnings surprises. However, the overly optimistic forecasts may lead to the undervaluation of earnings shocks, which amplifies the investors' sensitivity to earnings shocks. At the same time, due to a general short-selling restriction in China's stock market, analysts are required to furnish a higher number of "buy" rating reports to enhance commission income, thereby leading to a stronger optimistic bias [7, 8]. Thus, there exists a greater need for conducting an exhaustive analysis of the effectiveness of measuring earnings surprises in China's stock market.

Earnings surprises are widely used in the basic research on stock markets such as market efficiency and earning management. How to better measure earnings surprises is a core question in related empirical researches. The methods employed to measure earnings surprises continue to evolve. For instance, Brown et al. [9] illustrated that earnings surprises based on analysts' forecasts provided better predictions of corresponding abnormal returns than that

**Competing interests:** NO authors have competing interests.

derived from derived from time-series models' forecasts. Conversely, O'Brien [1] demonstrated that forecasts' errors based on the Foster Model proved to better explain abnormal returns than those established by analysts. Such as some financial valuation ratios can also be used to predict futures tock earnings [10–12]. Subsequently, most literature acknowledges analysts' earnings forecasts as a more reliable indication of stock market consensus and uses them as a proxy for market consensus expectations [13]. Furthermore, Bouchaud [14] uses the mean of analysts' forecasts as a proxy for market consensus, but Gu and Wu [15] based on minimizing the mean-squared error loss function, found that the median earnings forecasts are a better proxy for market consensus than the average. In addition, other academic studies suggest that the most recent earnings forecasts contain all public and private information, and are a better proxy for market consensus [16]. Nevertheless, Chiang et al., [3] point out that earnings surprises based on the median or the latest analysts' earnings forecasts still encounter systematic forecast bias, although they dilute the effect of outliers on the average. There exist certain restrictions in the literature. On one hand, it is worth noting that while current studies typically rely on analysts' earnings forecasts to estimate earnings surprises [17], the literature on earnings surprises measurement methods has primarily focused on comparing random walk models to analyst earnings forecasts, which lack of discussion on the different calculation methods of analysts' earnings forecasts; on the other hand, with the rapid development of China's capital market, leading to an increase in the number of analysts and greater attention paid to their opinions. Consequently, there is a need for further research on the validity of earnings surprises measurement in China's stock market.

How to test the accuracy of various earnings surprise measures? The mainstream literature employs event study method to investigate the relationship between earnings surprises and market reaction [3, 13]. They examine a noticeable positive correlation exists between earnings surprises and cumulative abnormal return (CAR) around the earnings announcement. However, some studies have revealed the substantial influence of financial anomalies on individual stock returns around earnings announcement. For example, Saver and Wilson [18] discovered that stocks which released their earnings announcements early and provided preliminary market cash flow information garnered a higher risk premium compared to those that disclosed their earnings at a later time. Similarly, Engelberg et al. [19] identified 97 distinct financial anomalies coinciding with earnings announcements, noting that the abnormal return during this period is 6 times higher than normal. Therefore, financial anomalies potentially introducing bias into CAR measurements. Thus, we believe that CAR needs to be amended to eliminate the effects of financial anomalies.

First, we examine whether CAR can accurately represent investors' reactions to earnings information. Due to the unique announcements disclosure mechanism in China, some companies are compelled to disclose the preliminary earnings by CSRC. About half of the listed companies opt to disclose the preliminary earnings before the end of February, then release the annual report around April. Although the preliminary earnings are not audited, they are usually seemed as an important reference to the actual earnings by investors with high accuracy [20]. Therefore, we calculate the difference between the preliminary earnings and the actual earnings as the earnings correction (EPS_PUB), which be the proxy for earnings surprises for specified firms. If EPS_PUB is zero, then there should be no earnings shock around annual announcement. So, we can apply EPS_PUB to assess the of validity of CAR. We find that only when there are earnings shocks, there is a significant positive correlation between EPS_PUB and CAR, but when there is no earnings shock, the correlation is insignificant, and the coefficient of EPS_PUB declines by about 80%. This indicates the stock return in the time window is dominated by other non-earnings factors when there is no earnings shock. According to the Efficient Market Hypothesis, stock prices follow a random walk and exhibit price

jumps in response to earnings shocks. Reference from Jiang and Zhu [21], we introduce an innovative revision to CAR, denoted as CAR_NEW, incorporating stock price jumps, and we validate that CAR_NEW is an effective proxy for investors' reactions to earnings correction (EPS_PUB).

Second, we explore the correlation between various measures of earnings surprises and revised market reaction (CAR_NEW). Most literature uses the analysts' forecast error (ERROR) as the market consensus [14, 22], while Chiang et al. [3] proposed a new measure of earnings surprises, that is the fraction of misses on the same side (FOM, the mean of analysts' earnings forecasts' signs). They find that FOM is the better measure of actual earnings surprises within the U.S. stock market. We find that ERROR is not significantly related to CAR_-NEW, whereas FOM is positively related to CAR and CAR_NEW. Then, when we put all measures of earnings surprise together as explanatory variables, only FOM remains significantly positive. These results indicate that FOM is a better proxy than other measures and absorbs the most unexpected earnings information reflected by ERROR. Moreover, earnings surprises calculated by star analysts, who should be with stronger abilities, still less effective than FOM. In addition, we also study the Post-Earnings Announcement Drifts (PEAD) with different measures of earnings surprises. When there is earnings shock, earnings surprises measured by different methods are significantly and positively correlated with post-earnings announcement cumulative abnormal return (POSTCAR), while there is no earnings shock, only FOM remains significantly positive. We further divide the samples of stock price jumps into positive jumps and negative jumps. We find that the systematic optimistic bias leads to analysts' overestimation of earnings, and earnings surprises estimated based on analysts' earnings forecasts are higher than market consensus, which amplifies the market reaction to earnings information, and also weakens the market reaction to earnings information when it is lower than market consensus.

In addition, we examine the validity of earnings surprises from other two aspects. On the one hand, if earnings surprises reflect the actual earnings shock for investors, there should exist a significantly positive relation between earnings surprises and investor trading behavior. Thus, we use high-frequency data to track the trading behavior of investors and examine whether earnings surprises are in line with investors trading behavior around earnings announcement. The result shows that only FOM exhibits a substantial positive correlation with order imbalance, i.e., investors show a preference for buying over selling when FOM is positive, which provides new evidence that FOM is a robust measure of earnings surprises. On the other hand, we investigate PEAD strategies for different earnings surprise and find that the premium of PEAD constructed by FOM can fully explain that by ERROR. These results also show that FOM is a more validity measure of earnings surprises.

Finally, we also carry out a series of robustness tests to ensure the reliability of our conclusion, including considering the impact of earnings pre-announcement, a more robust calculation of cumulative excess return, and other earnings surprises measurement methods.

The contribution of this paper may be reflected in the following aspects: First, we expand the literature related to earnings announcement and market reaction, previous literature has used the event study method to examine the relationship between earnings surprises and market reaction [3, 13]. However, they ignore the influence of financial anomalies on individual stock returns around the earnings announcement [19]. As far as we know, this is the first paper to point out such problem and prove that the market reaction measured by cumulative abnormal return cannot effectively represent the investors' response to earnings correction; Second, we build an appropriate metric for the actual investors' response to earnings shocks, i.e. revised market reaction (CAR_NEW), and verify that the revised market reaction can be a better measure of the investors' response to earnings correction; Third, we expand the

literature about the measurements of earning surprises and prove FOM is a better proxy for earnings surprises in China's stock market. We examine the validity of different earnings surprises measures based on analysts' earnings forecasts, and find that investors' behaviors are in line with FOM in China's stock market.

The structure of this paper is as follows: The first part is the introduction; The second part is hypothesis development; The third part is data; The fourth part is variables definition; The fifth part is empirical findings; The sixth part is the robustness test; The last part is the conclusion.

## 2. Hypothesis development

Some studies have revealed the substantial influence of financial anomalies on individual stock returns around earnings announcement. Thus, we believe that CAR needs to be amended to eliminate the effects of financial anomalies, and we examine whether CAR can accurately represent investors' reactions to earnings information.

Maheu and McCurdy [23] shows that the inflow of major information would have a significant influence on the volatility of stock returns. Therefore, we use the stock price jump around the earnings announcement to proxy for the earnings information shock. When the stock price jumps, it indicates that there is a major unexpected earnings information, and the earnings surprise is large; Conversely, when there is no jump, the stock price follows a random walk and there is no significantly unexpected earnings information. therefore, we use stock price jumps as a proxy for information shocks and construct corrective market reactions.

First, we examine whether CAR or CAR_NEW is a better proxy for investors' reactions to earnings information. Due to the unique announcements disclosure mechanism in China, we calculate the difference between the preliminary earnings and the actual earnings as the earnings correction (EPS_PUB), which be the proxy for earnings surprises for specified firms. If EPS_PUB is zero, then there should be no earnings shock around annual announcement. So, we can apply EPS_PUB to assess the of validity of CAR and CAR_NEW, and our hypothesis is stated as follows:

***H1*: *EPS_PUB is significantly related to CAR_NEW, and EPS_PUB is significantly related to CAR*.**

Chiang et al. [3] proposed a new measure of earnings surprises, that is the fraction of misses on the same side (FOM, the mean of analysts' earnings forecasts' signs). The earnings surprises calculated by the mean, the median or the latest of analysts' earnings forecasts all face the problem of systematic bias, the influence of outliers of FOM is small, which reduces the influence of extreme of analysts' forecasts error. They find that FOM is the better measure of actual earnings surprises within the U.S. stock market.

Due to short selling in the Chinese A-share market is substantially limited, for example, short selling was banned prior to 2010 and 2016 restrictions prohibited the short selling of nearly 70% of stocks. These short-selling restrictions [24]. Then analysts are required to furnish a higher number of "buy" rating reports to enhance commission income, thereby leading to a stronger optimistic bias [7]. Thereby FOM is a better proxy for earnings surprises in China's stock market, rather than the earnings surprises calculated by the mean, the median or the latest of analysts' earnings forecasts. Our second hypothesis is thus as follows:

***H2*: *FOM is significantly related to CAR_NEW, and ERROR is significantly related to CAR_NEW*.**

Then we examine the validity of earnings surprises from other two aspects. First, we use high-frequency data of investors' trading to capture investors' active behavior, examine whether investor trading behavior is significantly related to earnings surprises around earnings announcement. Then we employ a portfolio-based approach to test whether different measures of earnings surprises obtain difference PEAD. Specifically, we construct two groups, based on FOM and on ERROR, to evaluate their ability to explain each other's performance. Our third hypothesis is thus as follows:

**H3: Investor trading behavior is significantly related to FOM, and investor capture a more validity premium of PEAD constructed by FOM than ERROR.**

## 3. Data

We obtain daily stock return, analysts' earnings forecasts, earning announcement data and basic financial data for China A-shares from the China Stock Market and Accounting Research (CSMAR) database (https://data.csmar.com/). CSMAR is a comprehensive research-oriented database focusing on China Finance and Economy and based on academic research needs, meeting with the international professional standards while adapting to China's features. Our sample starts in January of 2001; the Chinese stock market is relatively undeveloped before then, and has more accuracy data of analysts' data. Our sample of earning announcements ends in June 2021, the analysts' earnings forecasts ends in April 2021 and the daily stock return ends in June 2021.

We sourced the earnings forecasts and basic financial data from the CSMAR database. As analysts predominantly forecast annual earnings for stocks, our analysis is limited to annual earnings data. We process the sample data as follows: (1) retain only the annual analysts' earnings forecasts made within 1 years of the annual announcement, (2) exclude samples with less than 5 analysts' earnings forecasts for each year; (3) exclude samples with less than 1 year of presence on the A -share; (4) exclude companies from the financial industry; (5) exclude samples with missing variable data. To mitigate the potential influence of outliers on our conclusions, we winsorized all core continuous variables at the 1st and 99th percentiles. Ultimately, our sample comprises 16,116 company-year observations.

## 4. Variables definition

### 4.1. Annual earnings correction (EPS_PUB)

Due to unique announcements disclosure mechanism in Chinese A-share stock market, we calculate the annual earnings correction (EPS_PUB) as the difference between the earnings reported in the preliminary earnings and those in the annual report:

$$EPS_{PUB_{j,t}} = \frac{EPS_{N_{j,t}} - EPS_{K_{j,t}}}{EPS_{N_{j,t}}} \tag{1}$$

where $EPS\_N_{j,t}$ is the earnings of annual report of stock j for year t, and $EPS\_K_{j,t}$ is the earnings of the preliminary earnings of stock j for year t.

### 4.2. Earnings surprises (S)

**4.2.1. Average of analysts' earnings errors (Akbas [25]).** earnings surprises are calculated based on the average of all analysts' earnings errors (ERROR1), the general calculation

formula is:

$$ERROR1_{j,t} = \frac{MEPS_{j,t} - FEPS_{j,t}}{MEPS_{j,t}} \tag{2}$$

where $FEPS_{j,t}$ is the average of all analysts' earnings forecasts for stock j in year t, $MEPS_{j,t}$ is the actual earnings of stock j for year t. When ERROR1>0, it indicates that analysts' forecast were lower than the actual earnings, resulting in positive earnings surprises. Conversely, when ERROR1<0, it indicates that analysts' earnings forecasts were less than the actual earnings.

**4.2.2. Latest analysts' earnings errors (Chiang et al., [3]).** We also calculate earnings surprises using the median of the latest analysts' earnings errors. These earnings surprises are denoted as ERROR2, replacing the average of analysts' earnings forecasts with the most recent analysts' earnings forecasts.

**4.2.3. Fraction of misses on the same side.** Following Chiang et al. [3], FOM is the average of the signs of analysts' earnings forecasts:

$$FOM_{j,t} = \frac{K_{j,t}}{N_{j,t}} - \frac{M_{j,t}}{N_{j,t}} \tag{3}$$

where $K_{j,t}$ is the number of forecasts that are lower than $MEPS_{j,t}$, and $M_{j,t}$ is the number of forecasts that are higher than $MEPS_{j,t}$, $N_{j,t}$ is the total number of the forecasts. FOM measures the fraction of misses on the same side.

This section may be divided by subheadings. It should provide a concise and precise description of the experimental results, their interpretation, as well as the experimental conclusions that can be drawn.

## 4.3. Revised market reaction (CAR_NEW)

Market reaction (CAR) is the cumulative excess return within three trading days around earnings announcement:

$$CAR_{j,t} = \prod_{k=1}^{T} \left(1 + R_{j,k} - R_{m,k}\right) - 1 \tag{4}$$

where $R_{j,k}$ is the raw return of stock j on day k of years t, $R_{m,k}$ is the market return weighted by the return of all A-share stocks on day k.

If EPS_PUB equals zero, it implies that there if no earnings shock in the annual report. We find that the average CAR around annual announcement is 0.39%, significantly different from zero, CAR cannot proxy for investors' reactions to earnings correction. Then in accordance with the Efficient Market Hypothesis, we construct the revised market reaction (CAR_NEW) based on stock price jumps.

Jiang and Zhu [21] demonstrated that stock prices experience jumps in response to significant information shocks, therefore, we use stock price jumps as a proxy for information shocks. The test statistics of stock price jumps are defined by Jiang and Oomen [26] and Jiang and Yao [27]:

$$\frac{V_{(0,T)}N}{\sqrt{\Omega_{SWV}}} \left(1 - \frac{RV_N}{SWV_N}\right)^d \to N(0,1) \tag{5}$$

If the stock price exhibits a positive jump during the annual earnings announcement (- 1,1), then JUMP = 1; If there is a negative jump in the stock price during the annual earnings announcement (- 1,1), then JUMP = -1; otherwise, JUMP = 0.

If there is a significant earnings surprises during the earnings announcement and the stock price jumps, CAR_NEW takes on a non-zero value. Conversely, when the stock price does not jump, the return follows a random walk during the earnings announcement, and CAR_NEW is set to zero:

$$CAR_{NEW} = \begin{cases} CAR, & Jump \neq 0 \\ 0, & Jump = 0 \end{cases} \tag{6}$$

## 4.4. Other control variables

Following Ran et al. [28–30], our control variables: (1) Company Size (Asset): the natural logarithm of the total assets in the current year; (2) Book to market ratio (BM): the ratio of book assets to market value; (3) Profitability (Roa): the return on assets; (4) Financial Leverage (Lev): the asset-liability ratio; (5) Institutional Shareholding(Inti): the institutional shareholding ratio at the end of the year; (6) Nature of Property (Nature): property rights, with 1 indicating state-owned enterprises and 0 indicating non-state-owned enterprises; (7) Time of Establishment of Company (Age): logarithm of the year of establishment of the stock.

## 4.5. Descriptive statistics

Table 1 shows the descriptive statistics results of the core variables. The mean of EPS_PUB is -0.016, signifying a variance between the preliminary earnings and the annual report earnings. Although the annual report revises the preliminary earnings, the overall revision range remains modest.

For the variable ERROR1, representing the average of analysts' earnings errors, the mean is -0.536, and the mean of ERROR2 is -0.155, the earnings surprises based on the latest value of analysts' forecasts is smaller with that based on the average. This trend suggests that analysts' forecasts tend to exhibit greater accuracy as the release of the annual report, the finding consistent with extant literature [16]. On average, the actual earnings are smaller than the analysts' earnings forecasts, consistent with the analysts' optimism bias [3, 15]. The mean of FOM is

**Table 1. Descriptive statistics.**

| Variables | N | MEAN | STD | P1 | P25 | MED | P75 | P99 |
|---|---|---|---|---|---|---|---|---|
| EPS_PUB | 7142 | -0.016 | 0.375 | -0.333 | -0.01 | 0 | 0 | 0.145 |
| FOM | 16116 | -0.394 | 0.651 | -1 | -1 | -0.657 | 0.071 | 1 |
| ERROR1 | 16116 | -0.536 | 5.278 | -9.077 | -0.41 | -0.122 | 0.007 | 1.597 |
| ERROR2 | 16116 | -0.155 | 2.061 | -3.353 | -0.109 | -0.004 | 0.027 | 0.940 |
| ERROR3 | 15860 | -0.01 | 0.027 | -0.12 | -0.012 | -0.004 | 0 | 0.025 |
| JUMP | 16116 | -0.021 | 0.33 | -1 | 0 | 0 | 0 | 1 |
| CAR | 16116 | 0.044 | 5.391 | -12.480 | -3.020 | -0.400 | 2.640 | 16.810 |
| CAR_NEW | 16116 | -0.048 | 2.437 | -9.060 | 0.000 | 0.000 | 0.000 | 9.470 |
| Asset | 16116 | 22.592 | 1.364 | 20.231 | 21.604 | 22.381 | 23.372 | 26.688 |
| BM | 16116 | 0.042 | 0.033 | 0.006 | 0.021 | 0.033 | 0.053 | 0.186 |
| Roa | 16116 | 0.086 | 0.058 | -0.058 | 0.047 | 0.076 | 0.113 | 0.292 |
| Lev | 16116 | 0.434 | 0.197 | 0.058 | 0.277 | 0.433 | 0.587 | 0.847 |
| Inti | 16116 | 50.962 | 24.814 | 1.522 | 32.25 | 55.373 | 70.59 | 93.206 |
| Nature | 16116 | 0.404 | 0.491 | 0 | 0 | 0 | 1 | 1 |
| Age | 16116 | 2.058 | 0.776 | 0 | 1.386 | 2.197 | 2.708 | 3.258 |

-0.394, and the standard deviation is 0.651, which is less than the standard deviation of ERROR1 and ERROR2, indicating that FOM is less influenced by extreme values. Within our sample, 3.22% exhibit positive jumps, while 5.04% record negative jumps.

## 5. Empirical analysis

### 5.1. Whether the market reaction can effectively proxy investors response

In this section, we use the unique performance disclosure mechanism of China's stock market, select companies that publish both preliminary earnings and annual report, then we test whether CAR can effectively proxy for investors' reactions to earnings correction. While preliminary earnings constitute the primary unaudited operational data, their reliability remains consistently robust. From the descriptive statistics in Table 1, the difference between preliminary earnings and annual report is almost zero from the 25th percentile to the 75th percentile, and only 1/4 of the samples have differences, with the annual earnings correction (EPS_PUB) deviating by a mere 3% from the mean of analysts' forecast errors. Therefore, after the release of the preliminary earnings, investors take the earnings released by preliminary earnings as the expectation of the actual earnings. Subsequently, upon the issuance of annual report, EPS_PUB is a reasonable proxy for earnings surprises [20]. We test whether it can proxy the investors' reactions to earnings correction during the earnings announcement.

We use the stock price jumps to describe the information shocks during earnings announcement, and divides stocks into two groups according to whether the stock price jumps, that is, the stock price jumps (JUMP≠0), and the stock price no jump (JUMP = 0). Regression of two different samples as follow:

$$CAR_{j,t} = \alpha + \gamma \text{EPS}_{\text{PUB}_{j,t}} + Controls_{j,t} + \varepsilon_{i,j,t} \tag{7}$$

where CAR is the market reaction, the cumulative excess return during the earnings announcement, EPS_PUB is the annual earnings correction, and Controls are control variable.

The results are shown in Table 2, while the stock price jumps during the earnings announcement (JUMP≠0), the coefficient of the annual earnings correction (EPS_PUB) is 0.675, which is statistically significant at 1%, After adding control variables, the significance remains unchanged. In the sample that the stock price no jump (JUMP = 0), that is, the stock price follows random walk in the earnings announcement, and the relationship between CAR and EPS_PUB is not significant, the t-value is only 1.20. After controlling other variables, the relation between CAR and EPS_PUB is still not significant. The coefficient of EPS_PUB is about 80% lower than the sample in which the stock price jumps.

Only in the presence of an earnings information shock, EPS_PUB is positively correlated with the CAR, and the CAR can act as a proxy for the investor response. When there is no earnings information shock in the window, CAR may be interfered by financial anomalies, and the measurement of investors' response is biased, which cannot effective proxy for investors' reactions to earnings correction. It shows that when there is no earnings information shock in annual report announcement, the CAR is disturbed by other non-earnings information factors. Therefore, we construct revised market reaction (CAR_NEW).

### 5.2. Test revised market reaction

In this section, we examine whether the revised market reaction (CAR_NEW) enhances the market reaction (CAR). We perform regressions of both CAR and CAR_NEW on the annual

**Table 2. CAR and EPS_PUB: Stock price jumps.**

| | JUMP≠0 | JUMP≠0 | JUMP = 0 | JUMP = 0 |
|---|---|---|---|---|
| | CAR | CAR | CAR | CAR |
| EPS_PUB | 0.675*** | 0.603*** | 0.149 | 0.112 |
| | (3.83) | (2.97) | (1.20) | (0.90) |
| Asset | | 0.221 | | 0.388*** |
| | | (0.52) | | (4.37) |
| BM | | -60.039*** | | -26.787*** |
| | | (-2.83) | | (-7.26) |
| Roa | | -1.222 | | 1.021 |
| | | (-0.16) | | (0.74) |
| Lev | | -1.930 | | -0.656 |
| | | (-0.77) | | (-1.36) |
| Inti | | 0.022 | | 0.001 |
| | | (1.49) | | (0.39) |
| Nature | | 1.285 | | -0.033 |
| | | (1.46) | | (-0.18) |
| Age | | 0.120 | | -0.006 |
| | | (1.61) | | (-0.39) |
| Industry | YES | YES | YES | YES |
| Year | YES | YES | YES | YES |
| Constant | -0.519 | -4.815 | -1.088* | -8.816*** |
| | (-0.12) | (-0.51) | (-1.74) | (-4.87) |
| N | 545 | 545 | 6597 | 6597 |
| R2 | 0.193 | 0.221 | 0.011 | 0.022 |

T values in parentheses,

*, **, *** indicate significant at the 10%, 5%, and 1% levels, respectively. Same below. Regressions control for a fixed effect on both industries and years.

earnings correction (EPS_PUB) as follows:

$$CAR_{j,t}\left(CAR_{NEW_{j,t}}\right) = \alpha + \gamma EPS_{PUB_{j,t}} + Controls_{j,t} + \varepsilon_{i,j,t} \tag{8}$$

where $Controls_{j,t}$ is a vector of the firm-level control variables.

Table 3 shows the results for the two dependent variables. There is a statistically significant positive relationship between CAR_NEW and EPS_PUB, but no correlation is observed between CAR and EPS_PUB, suggesting that CAR_NEW effectively corrects CAR influenced by financial anomalies. Therefore, CAR_NEW seems to be a better proxy for the investors' reactions to earnings correction.

## 5.3. Effectiveness of different earnings surprises measures

**5.3.1. Different earnings surprises measures and (revised) market reaction.** We proceed to examine the relationship between various measures of earnings surprises and the (revised) market reaction. We aim to assess the validity of these different earnings surprises, including those derived from the average of analysts' earnings forecasts (ERROR1), the most recent analysts' earnings forecasts (ERROR2), and the fraction of misses on the same side (FOM) within the Chinese stock market.

**Table 3. Annual earnings correction and revised market reaction.**

|  | CAR | CAR | CAR_NEW | CAR_NEW |
|---|---|---|---|---|
| **EPS_PUB** | **0.271** | **0.227** | **0.164**** | **0.158**** |
|  | **(1.64)** | **(1.42)** | **(2.30)** | **(2.22)** |
| Asset |  | 0.369*** |  | 0.018 |
|  |  | (3.99) |  | (0.46) |
| BM |  | -30.545*** |  | -5.895*** |
|  |  | (-8.13) |  | (-3.57) |
| Roa |  | 0.809 |  | -0.120 |
|  |  | (0.58) |  | (-0.22) |
| Lev |  | -0.744 |  | -0.147 |
|  |  | (-1.52) |  | (-0.76) |
| Inti |  | 0.003 |  | 0.002* |
|  |  | (1.05) |  | (1.66) |
| Nature |  | 0.053 |  | 0.090 |
|  |  | (0.29) |  | (1.10) |
| Age |  | 0.003 |  | 0.009 |
|  |  | (0.22) |  | (1.36) |
| Industry | YES | YES | YES | YES |
| Year | YES | YES | YES | YES |
| Constant | -1.192** | -8.524*** | -0.242 | -0.599 |
|  | (-2.08) | (-4.57) | (-0.99) | (-0.75) |
| N | 7142 | 7142 | 7142 | 7142 |
| R2 | 0.016 | 0.027 | 0.018 | 0.021 |

This table reports results regarding the relation between CAR (CAR_NEW) and EPS_PUB, regressions control for a fixed effect on both industries and years. T values in parentheses,

*, **, *** indicate significant at the 10%, 5%, and 1% levels, respectively.

Therefore, we follow the regression:

$$CAR_{j,t}\left(CAR_{NEW_{j,t}}\right) = \alpha + \beta \cdot S_{j,t} + Controls_{j,t} + \varepsilon_{j,t} \tag{9}$$

where $S_{j,t}$ is the different measures of earnings surprises. The regression results are shown in Table 4. When employing CAR as the dependent variable, the regression coefficients for ERROR1, ERROR2 and FOM all exhibit significantly positive, these findings are consistent with Fried and Givoly [31], ERROR is a suitable proxy for earnings surprises, but the significance of the coefficient for FOM is 14.28, which is higher than the t-value of the ERROR1 (4.00) and the t-value of the ERROR2 (3.09). However, after controlling for the three earnings surprises measures, only the coefficient of FOM remains significantly positive in relation to CAR. In contrast, the coefficients of the other two earnings surprises become statistically insignificant. When using CAR_NEW as the dependent variable, no significant correlation is observed with ERROR1 and ERROR2, while only FOM retains its significant, the coefficient between FOM and CAR_NEW is 0.186 with a t-value of 5.59, which is significant at the 1% statistical level. These results suggest that FOM effectively captures unexpected earnings information present in ERROR1 and ERROR2, so it is the sole proxy for market earnings surprises.

The findings presented in Table 4 suggest that FOM fully absorbs the unexpected earnings information manifested in both ERROR1 and ERROR2. When contrasted with the prevalent method of using analysts' absolute earnings forecasts to calculate ERROR1 and ERROR2, the

**Table 4. Earnings surprises and market reaction.**

|  | CAR | CAR | CAR | CAR | CAR_NEW | CAR_NEW | CAR_NEW | CAR_NEW |
|---|---|---|---|---|---|---|---|---|
| **ERROR1** | 0.019*** |  |  | 0.001 | 0.003 |  |  | -0.013 |
|  | (4.00) |  |  | (0.05) | (1.41) |  |  | (-1.05) |
| **ERROR2** |  | 0.056*** |  | 0.032 |  | 0.021 |  | 0.044 |
|  |  | (3.09) |  | (0.63) |  | (1.22) |  | (1.17) |
| **FOM** |  |  | 1.037*** | 1.028*** |  |  | 0.186*** | 0.182*** |
|  |  |  | (14.28) | (14.13) |  |  | (5.59) | (5.46) |
| Asset | 0.300*** | 0.299*** | 0.256*** | 0.257*** | 0.070*** | 0.070*** | 0.062** | 0.062** |
|  | (5.47) | (5.46) | (4.72) | (4.75) | (2.88) | (2.90) | (2.56) | (2.55) |
| BM | -17.427*** | -17.410*** | -16.526*** | -16.544*** | -4.169*** | -4.170*** | -4.008*** | -4.001*** |
|  | (-9.03) | (-9.03) | (-8.76) | (-8.77) | (-4.94) | (-4.94) | (-4.76) | (-4.75) |
| Roa | 0.845 | 0.877 | -2.446*** | -2.484*** | -0.583 | -0.603 | -1.177*** | -1.156*** |
|  | (0.91) | (0.95) | (-2.58) | (-2.61) | (-1.37) | (-1.41) | (-2.65) | (-2.60) |
| Lev | -0.410 | -0.408 | -0.592* | -0.592* | -0.215 | -0.216 | -0.248* | -0.246* |
|  | (-1.25) | (-1.24) | (-1.82) | (-1.82) | (-1.51) | (-1.52) | (-1.74) | (-1.73) |
| Inti | 0.006** | 0.006** | 0.005** | 0.005** | 0.004*** | 0.004*** | 0.004*** | 0.004*** |
| Nature | -0.136 | -0.136 | -0.168 | -0.168 | -0.003 | -0.003 | -0.009 | -0.009 |
|  | (-1.18) | (-1.18) | (-1.47) | (-1.47) | (-0.07) | (-0.07) | (-0.19) | (-0.19) |
| Age | -0.002 | -0.002 | -0.002 | -0.002 | -0.001 | -0.001 | -0.001 | -0.001 |
|  | (-0.20) | (-0.19) | (-0.22) | (-0.23) | (-0.15) | (-0.16) | (-0.16) | (-0.14) |
| Industry | YES | YES | YES | YES | YES | YES | YES | YES |
| Year | YES | YES | YES | YES | YES | YES | YES | YES |
| Constant | -7.338*** | -7.330*** | -5.850*** | -5.875*** | -1.846*** | -1.850*** | -1.581*** | -1.582*** |
|  | (-5.58) | (-5.57) | (-4.48) | (-4.50) | (-3.10) | (-3.10) | (-2.64) | (-2.64) |
| N | 16116 | 16116 | 16116 | 16116 | 16116 | 16116 | 16116 | 16116 |

This table reports results regarding the relation between CAR (CAR_NEW) and different earnings surprises measures, regressions control for a fixed effect on both industries and years. T values in parentheses,

*, **, *** indicate significant at the 10%, 5%, and 1% levels, respectively.

FOM derived from the sign of analysts' earnings forecasts is a better proxy for market earnings surprises. The regression coefficients for earnings surprises are notably positive when each of the three earnings surprises measures are individually examined using CAR as an indicator of investors' reactions to earnings information. Each earnings surprises measurement appears effective, the existing literature often does not differentiate between these three measurements in their detailed examinations of earnings surprises.

However, after revising CAR, the regressions of ERROR1, ERROR2 and CAR_NEW are not significant, and only FOM remains significantly positive, indicating that the mean (the latest) of analysts' forecasts errors does not proxy for market consensus and that the fraction of misses on the same side is a better proxy for earnings surprises. Given the inherent biases in CAR, the results are debatable when CAR is employed to assess the validity of various earnings surprises measures. Neither ERROR1 nor ERROR2 can accurately proxy for earnings surprises, and the fraction of misses on the same side (FOM) is a better proxy for earnings surprises.

Table 4 shows that the fraction of misses on the same side outperforms earnings surprises calculated based on the mean (the latest) of analysts' forecasts errors. Some studies have shown that there is heterogeneity among analysts, star analysts demonstrate superior personal and information-processing capabilities compared to non-star analysts [32]. Consequently, the

earnings forecasts disseminated by star analysts tend to be more precise. Given this differential in analyst ability, we only select the sample of analysts' earnings forecasts after becoming star analysts, constructs earnings surprises (ERROR1_STAR, ERROR2_STAR), based on the mean (the latest) of star analysts' forecasts, and tests the validity of three different earnings surprises estimated by ERROR1_STAR, ERROR2_STAR and FOM. To focus on this narrowed sample, we include only companies that have earnings forecasts from star analysts, and FOM is calculated using the earnings forecasts from all analysts.

The results are shown in Table 5, are consistent with Table 4, the regression coefficients are significantly positive when the more capable star analysts replace the earnings forecasts of all analysts, regression ERROR1_STAR, ERROR2_STAR and FOM are all significantly positive with CAR, respectively, and only the coefficient of FOM is significantly positive when controlling for all three kinds of earnings surprises measures simultaneously; when replacing the explanatory variables with CAR_NEW, only the coefficient of FOM is significantly positive, and neither the coefficient of ERROR1_STAR nor ERROR2_STAR is significant. The conclusions in Table 5 show that even the star analysts with stronger individual ability, the mean (the latest) of their earnings surprises are still not as effective as FOM. Considering that FOM can covers more companies, using FOM as a proxy for earnings surprises is better than those calculated based on the mean (the latest) of star analysts' errors.

**Table 5. Star analysts.**

| | CAR | CAR | CAR | CAR | CAR_NEW | CAR_NEW | CAR_NEW | CAR_NEW |
|---|---|---|---|---|---|---|---|---|
| ERROR1_STAR | 0.017*** | | | 0.019 | 0.001 | | | -0.000 |
| | (5.39) | | | (0.73) | (0.91) | | | (-0.01) |
| ERROR2_STAR | | 0.031*** | | -0.015 | | 0.002 | | 0.000 |
| | | (4.66) | | (-0.31) | | (0.80) | | (0.02) |
| FOM | | | 1.259*** | 1.253*** | | | 0.171*** | 0.171*** |
| | | | (11.70) | (11.64) | | | (3.27) | (3.26) |
| Asset | 0.234*** | 0.232*** | 0.175** | 0.179** | 0.046 | 0.046 | 0.039 | 0.039 |
| | (3.10) | (3.08) | (2.38) | (2.44) | (1.46) | (1.46) | (1.22) | (1.22) |
| BM | -13.595*** | -13.547*** | -12.716*** | -12.783*** | -2.879** | -2.877** | -2.765** | -2.765** |
| | (-5.30) | (-5.28) | (-5.16) | (-5.18) | (-2.52) | (-2.52) | (-2.42) | (-2.42) |
| Roa | 3.242** | 3.225** | -0.667 | -0.732 | 0.235 | 0.232 | -0.309 | -0.309 |
| | (2.46) | (2.45) | (-0.50) | (-0.55) | (0.37) | (0.37) | (-0.46) | (-0.46) |
| Lev | 0.013 | 0.015 | -0.352 | -0.357 | -0.135 | -0.135 | -0.186 | -0.186 |
| | (0.03) | (0.03) | (-0.74) | (-0.75) | (-0.64) | (-0.64) | (-0.87) | (-0.87) |
| Inti | 0.007** | 0.007** | 0.006* | 0.006* | 0.004*** | 0.004*** | 0.004*** | 0.004*** |
| | (2.14) | (2.13) | (1.87) | (1.87) | (2.83) | (2.83) | (2.73) | (2.73) |
| Nature | -0.123 | -0.123 | -0.129 | -0.128 | -0.003 | -0.003 | -0.004 | -0.004 |
| | (-0.72) | (-0.73) | (-0.78) | (-0.77) | (-0.04) | (-0.04) | (-0.05) | (-0.05) |
| Age | -0.006 | -0.006 | 0.175** | 0.179** | 0.002 | 0.002 | 0.039 | 0.039 |
| | (-0.52) | (-0.52) | (2.38) | (2.44) | (0.37) | (0.36) | (1.22) | (1.22) |
| Industry | YES | YES | YES | YES | YES | YES | YES | YES |
| Year | YES | YES | YES | YES | YES | YES | YES | YES |
| Constant | -6.624*** | -6.576*** | -4.346** | -4.423** | -1.190 | -1.188 | -0.886 | -0.886 |
| | (-3.59) | (-3.57) | (-2.43) | (-2.48) | (-1.46) | (-1.46) | (-1.09) | (-1.08) |
| N | 7119 | 7119 | 7119 | 7119 | 7119 | 7119 | 7119 | 7119 |
| R2 | 0.028 | 0.028 | 0.048 | 0.049 | 0.016 | 0.016 | 0.018 | 0.018 |

T values in parentheses,

*, **, *** indicate significant at the 10%, 5%, and 1% levels, respectively.

**5.3.2. Different earnings surprises measures and Post-Earnings Announcement Drift.**
Brown [33] first discovered the financial anomaly known as the Post-Earnings Announcement Drift. Within this phenomenon, stocks demonstrating the most pronounced earnings surprises continue to ascend, while those with the weakest earnings surprises persistently decline post-announcement. This intriguing trend has been consistently reaffirmed in subsequent research [34–36]. Therefore, we further investigate the phenomenon of the Post-Earnings Announcement Drift caused by different earnings surprise measures, using the post-earnings announcement cumulative excess return (POSTCAR) to test the Post-Earnings Announcement Drift of three kinds of earnings surprises, we divided them into two samples according to whether the stock price jumps.

Table 6 shows our empirical findings. When JUMP≠0, the coefficient of ERROR1 and POSTCAR is 0.659, achieving significance at the 5% statistical level. This denotes the sensitivity of stock prices to earnings information shocks. Conversely, in instances where JUMP = 0, indicating an absence of stock price jumps and a potential adherence to a random walk model, the coefficient of ERROR1 and POSTCAR is a mere 0.041, lacking statistical significance.

Regardless of stock price jumps, the coefficients associated with ERROR2 and POSTCAR are statistically insignificant. Conversely, FOM exhibits a significant PEAD irrespective of the

**Table 6. Post-Earnings Announcement Drift: Whatever stock price jumps.**

| | JUMP≠0 | | | JUMP = 0 | | |
|---|---|---|---|---|---|---|
| | POSTCAR | POSTCAR | POSTCAR | POSTCAR | POSTCAR | POSTCAR |
| ERROR1 | 0.659** | | | 0.041 | | |
| | (2.03) | | | (1.25) | | |
| ERROR2 | | 0.467 | | | 0.156 | |
| | | (1.59) | | | (1.51) | |
| FOM | | | 6.387*** | | | 2.610*** |
| | | | (5.85) | | | (9.64) |
| Asset | -0.410 | -0.452 | -0.513 | -0.729*** | -0.728*** | -0.844*** |
| | (-0.47) | (-0.51) | (-0.59) | (-3.46) | (-3.45) | (-3.97) |
| BM | -103.663*** | -102.951*** | -106.901*** | -24.261*** | -24.260*** | -21.855*** |
| | (-4.01) | (-3.98) | (-4.16) | (-3.25) | (-3.25) | (-2.93) |
| Roa | -1.403 | 2.317 | -17.378 | -1.156 | -1.168 | -9.595*** |
| | (-0.10) | (0.17) | (-1.22) | (-0.33) | (-0.34) | (-2.65) |
| Lev | -3.941 | -3.897 | -5.216 | -3.155** | -3.150** | -3.622*** |
| | (-0.81) | (-0.80) | (-1.09) | (-2.55) | (-2.55) | (-2.92) |
| Inti | 0.052 | 0.056* | 0.047 | 0.024*** | 0.024*** | 0.022** |
| | (1.56) | (1.66) | (1.42) | (2.73) | (2.72) | (2.47) |
| Nature | 1.891 | 1.758 | 1.748 | -0.661 | -0.662 | -0.751* |
| | (1.15) | (1.07) | (1.09) | (-1.55) | (-1.55) | (-1.75) |
| Age | -0.055 | -0.047 | -0.066 | -0.074** | -0.074** | -0.074** |
| | (-0.50) | (-0.43) | (-0.61) | (-2.40) | (-2.39) | (-2.41) |
| Industry | YES | YES | YES | YES | YES | YES |
| Year | YES | YES | YES | YES | YES | YES |
| Constant | -5.002 | -5.049 | 1.535 | 20.685*** | 20.663*** | 24.449*** |
| | (-0.24) | (-0.24) | (0.08) | (4.15) | (4.14) | (4.91) |
| N | 1331 | 1331 | 1331 | 14785 | 14785 | 14785 |
| R2 | 0.150 | 0.149 | 0.171 | 0.075 | 0.075 | 0.081 |

T values in parentheses,

*, **, *** indicate significant at the 10%, 5%, and 1% levels, respectively.

presence or absence of stock price jumps during earnings announcements. Specifically, with JUMP≠0, the coefficient of FOM is 6.382—nearly three times its counterpart (2.598) in cases devoid of stock price jumps. This suggests the PEAD is larger in stocks with significant earnings information shocks in the market. Moreover, the t-value for FOM is 5.85, which is larger than the t-value of the ERROR1, 2.03. This finding also suggests that whether there is a significant earnings information shock, the fraction of misses on the same side is better proxy for earnings surprises.

We further divide the sample of stock price jumps (JUMP≠0) into positive jumps (JUMP>0) and negative jumps (JUMP<0). Table 7 shows the results of this subdivision. For positive jumps (JUMP>0), where EPS is higher than expected and both ERROR1 and FOM coefficients with POSTCAR are significantly positive. Notably, the coefficient for FOM is 9.84, approximately 50% elevated compared to its value for JUMP≠0 as presented in Table 7. On the other hand, when considering negative jumps (JUMP<0), all three earnings surprise coefficients relative to POSTCAR are statistically inconsequential. These observations resonate with prior literature highlighting analysts' systematic optimism bias [1–3]. Such bias suggests analysts tend to overestimate EPS, resulting in inflated earnings

**Table 7. PEAD: Whatever stock price jumps.**

| | JUMP>0 | | | JUMP<0 | | |
|---|---|---|---|---|---|---|
| | POSTCAR | POSTCAR | POSTCAR | POSTCAR | POSTCAR | POSTCAR |
| ERROR1 | 2.262* | | | -0.108 | | |
| | (1.85) | | | (-0.31) | | |
| ERROR2 | | 0.255 | | | 0.501 | |
| | | (1.33) | | | (0.87) | |
| FOM | | | 9.840*** | | | 1.918 |
| | | | (5.13) | | | (1.46) |
| Asset | -0.827 | -0.699 | -1.041 | -0.252 | -0.285 | -0.336 |
| | (-0.56) | (-0.47) | (-0.72) | (-0.31) | (-0.35) | (-0.41) |
| BM | -129.142*** | -130.436*** | -144.407*** | -17.842 | -17.696 | -16.100 |
| | (-2.97) | (-2.97) | (-3.30) | (-0.67) | (-0.66) | (-0.60) |
| Roa | 19.613 | 27.591 | -3.303 | -1.317 | -3.229 | -7.980 |
| | (0.87) | (1.23) | (-0.15) | (-0.08) | (-0.21) | (-0.50) |
| Lev | -0.350 | -0.249 | -3.289 | -7.057 | -7.080 | -7.360 |
| | (-0.04) | (-0.03) | (-0.39) | (-1.37) | (-1.37) | (-1.42) |
| Inti | -0.076 | -0.076 | -0.085 | 0.041 | 0.041 | 0.042 |
| | (-1.15) | (-1.14) | (-1.31) | (1.10) | (1.08) | (1.12) |
| Nature | 4.267 | 4.498 | 4.717 | 0.038 | 0.204 | 0.093 |
| | (1.31) | (1.38) | (1.49) | (0.02) | (0.13) | (0.06) |
| Age | 0.034 | 0.039 | 0.008 | -0.171 | -0.167 | -0.174 |
| | (0.18) | (0.20) | (0.04) | (-1.53) | (-1.49) | (-1.57) |
| Industry | YES | YES | YES | YES | YES | YES |
| Year | YES | YES | YES | YES | YES | YES |
| Constant | 41.128 | 38.017 | 51.003* | -7.853 | -6.507 | -3.508 |
| | (1.36) | (1.24) | (1.70) | (-0.42) | (-0.35) | (-0.18) |
| N | 519 | 519 | 519 | 812 | 812 | 812 |
| R2 | 0.068 | 0.062 | 0.107 | 0.040 | 0.041 | 0.043 |

T values in parentheses,

*, **, *** indicate significant at the 10%, 5%, and 1% levels, respectively.

surprises based on their forecasts. This can accentuate market reactions when actual earnings surpass market consensus, yet attenuate responses when they underperform against the same benchmark.

## 5.4. Further inspiration: Order imbalance

Our results suggest that FOM captures a more comprehensive set of earnings-related information. To further substantiate our conclusions, we employ high-frequency trading data from investors to track their trading behavior. Following Kelley and Tetlock [37], we calculate investor order imbalance within three trading days around earnings announcement. The share-weighted imbalance measure is a natural way to aggregate investors' opinions about a particular stock:

$$Imb = \frac{shares\ bought - shares\ sold}{(shares\ bough + shares\ sold)} \tag{10}$$

subsequently, we segment Imb into Imb_big and Imb_small based on order size. Then, we follow the regression:

$$Imb_{j,t} = \alpha + \beta \cdot S_{j,t} + Controls_{j,t} + \varepsilon_{j,t} \tag{11}$$

Table 8 presents a significant positive correlation between FOM and investors' active behavior. Interestingly, the evidence shows that investors usually trade using FOM rather than ERROR. The similar results are also found in subsample, thus indicating that FOM is a better proxy for earnings surprises.

## 5.5. Different earnings surprises measures: PEAD

We employ a portfolio-based approach to test whether different measures of earnings surprises obtain difference PEAD. Our analysis centers on exploring the variations in average returns that can be elucidated by each measure. Specifically, we construct two groups, based on FOM and on ERROR1, to evaluate their ability to explain each other's performance.

Our methodology can be summarized as follows: In each year, stocks are allocated to five groups using FOM, and are calculated the cumulative excess return after the earnings announcement (2, 61). PEAD_FOM is the difference between the value-weighted average of the returns on the high-FOM portfolio and the average of the returns on the low-FOM portfolio. In the same way, stocks are allocated to five groups using ERROR1 in every year, and are calculated the portfolio return (PEAD_ERROR1) of PEAD. Then e_mkt and v_mkt are the equal-weighted and value-weighted excess portfolio returns adjusted by market return. e_ff3 and v_ff3 are the equal-weighted and value-weighted excess portfolio returns adjusted by the FF3 factors, respectively.

Table 9 shows that PEAD_FOM is capable of explaining PEAD_ERROR1, implying that FOM reflects more earnings surprises information. When investors make their investment decisions on analysts' earnings forecasts, the PEAD strategy constructed by FOM achieves higher return, suggesting that FOM can be a better proxy for earnings surprises.

## 6. Robustness test

### 6.1. Eliminate the effect of earnings pre-announcement and preliminary earnings

Due to the unique performance disclosure mechanism of China's stock market, some companies will release the earnings pre-announcement or the preliminary earnings, which precede

**Table 8. Further inspiration: order imbalance.**

|  | Imb | Imb | Imb | Imb | Imb_big | Imb_small |
|---|---|---|---|---|---|---|
| **ERROR1** | **0.000** |  |  | **-0.000** | **-0.000** | **-0.000** |
|  | **(1.51)** |  |  | **(-0.31)** | **(-0.43)** | **(-0.19)** |
| **ERROR2** |  | **0.000** |  | **0.000** | **0.001** | **0.001** |
|  |  | **(1.22)** |  | **(0.30)** | **(0.61)** | **(0.92)** |
| **FOM** |  |  | **0.015*** | **0.015*** | **0.018*** | **0.011*** |
|  |  |  | **(11.75)** | **(11.72)** | **(9.36)** | **(9.17)** |
| Asset | 0.015*** | 0.015*** | 0.014*** | 0.014*** | 0.024*** | 0.018*** |
|  | (14.40) | (14.41) | (13.94) | (13.93) | (14.37) | (16.92) |
| BM | -0.433*** | -0.433*** | -0.420*** | -0.420*** | -0.714*** | -0.419*** |
|  | (-10.38) | (-10.38) | (-10.15) | (-10.15) | (-10.63) | (-10.19) |
| Roa | 0.061*** | 0.061*** | 0.013 | 0.013 | 0.014 | 0.055*** |
|  | (4.01) | (4.02) | (0.79) | (0.80) | (0.58) | (3.62) |
| Lev | -0.015*** | -0.015*** | -0.018*** | -0.018*** | -0.019** | -0.017*** |
|  | (-2.60) | (-2.60) | (-3.07) | (-3.07) | (-1.97) | (-2.98) |
| Inti | -0.000 | -0.000 | -0.000 | -0.000 | -0.000 | -0.000*** |
|  | (-0.55) | (-0.54) | (-0.88) | (-0.88) | (-1.31) | (-3.00) |
| Nature | 0.000 | 0.000 | -0.000 | -0.000 | 0.005 | -0.003 |
|  | (0.06) | (0.06) | (-0.16) | (-0.16) | (1.46) | (-1.36) |
| Age | 0.000 | 0.000 | 0.000 | 0.000 | -0.000 | 0.001*** |
|  | (1.40) | (1.40) | (1.38) | (1.38) | (-0.19) | (4.97) |
| Industry | YES | YES | YES | YES | YES | YES |
| Year | YES | YES | YES | YES | YES | YES |
| Constant | -0.238*** | -0.238*** | -0.217*** | -0.217*** | -0.363*** | -0.346*** |
|  | (-7.31) | (-7.31) | (-6.58) | (-6.58) | (-8.40) | (-8.78) |
| N | 16042 | 16042 | 16042 | 16042 | 15994 | 16041 |
| R2 | 0.167 | 0.167 | 0.174 | 0.174 | 0.167 | 0.168 |

This table reports results regarding the relation between order imbalance and different earnings surprises measures. The order imbalance measured using shares bought minus shares sold divided by shares bought plus shares sold. Imb using the full order flows, Imb_big is calculated when the amount is greater than 200,000, Imb_small is calculated when the amount is smaller than 200,000. T values in parentheses,

*, **, *** indicate significant at the 10%, 5%, and 1% levels, respectively.

the annual reports. earnings pre-announcement typically provides a generalized earnings range, thereby offering a coarser granularity of earnings data [28]. In contrast, preliminary earnings convey more precise information. In order to avoid the early reaction of earnings information, we delete stocks that issued earnings pre-announcement within the year. For firms with preliminary earnings, the publication date of these estimates is designated as the earnings announcement date.

The results in Table 10 are adjusted samples. The initial four columns, with market reaction as the dependent variable, reveal that the coefficients of ERROR1, ERROR2 and FOM are significantly positive. Conversely, the subsequent quartet of columns employs a revised market reaction as the dependent variable. The results are consistent with the previous section, only the coefficient of FOM is significantly positive, the coefficients of ERROR1 and ERROR2 are insignificant, suggesting FOM is a more accurate measurement of earnings surprises. While CAR struggles to discern the optimal earnings surprise estimate, CAR_-NEW reinforces the assertion that FOM serves as the most efficient proxy for earnings surprises.

**Table 9. Relationship of PEAD_FOM and PEAD_ERROR1.**

**Panel A. $PEAD\_FOM_t = \alpha + \beta \cdot PEAD\_ERROR1_{1t} + \varepsilon_t$**

| | PEAD_FOM | | | |
|---|---|---|---|---|
| | e_mkt | v_mkt | e_ff3 | v_ff3 |
| PEAD_ERROR1 | 0.972*** | 0.770*** | 0.916*** | 0.540** |
| | (10.09) | (4.75) | (5.44) | (2.27) |
| $\alpha$ | **1.532**** | **1.889** | **1.881*** | **3.078**** |
| | **(2.51)** | **(1.68)** | **(2.07)** | **(2.15)** |
| N | 19 | 19 | 19 | 19 |
| R2 | 0.857 | 0.570 | 0.635 | 0.233 |

**Panel B. $PEAD\_ERROR1_t = \alpha + \beta \cdot PEAD\_FOM_{1t} + \varepsilon_t$**

| | PEAD_ ERROR1 | | | |
|---|---|---|---|---|
| | e_mkt | v_mkt | e_ff3 | v_ff3 |
| PEAD_ FOM | 0.881*** | 0.741*** | 0.694*** | 0.432** |
| | (10.09) | (4.75) | (5.44) | (2.27) |
| $\alpha$ | **-0.732** | **0.647** | **0.175** | **2.040** |
| | **(-1.12)** | **(0.55)** | **(0.20)** | **(1.50)** |
| N | 19 | 19 | 19 | 19 |
| R2 | 0.857 | 0.570 | 0.635 | 0.233 |

Each year, stocks are allocated to five groups using FOM and ERROR1 respectively, each year, calculate the difference between the value-weighted average of the returns on the high portfolio and the average of the returns on the low portfolio as PEAD_FOM and PEAD_ERROR1, respectively. T values in parentheses,

*, **, *** indicate significant at the 10%, 5%, and 1% levels, respectively.

Since both ERROR1 and ERROR2 are calculated using the analysts' forecast absolute value, and the results of both are similar, so the following test only shows the results of ERROR1.

## 6.2. Earnings surprises test: Stock price jumps and revised market reaction adjustment

The previous section uses stock price jumps (JUMP) to describe earnings information shocks, if a stock has a significant unexpected earning, its stock price should jump during the earnings announcement. Although JUMP only measures whether the stock price jumps, it does not quantify the magnitude of earnings surprises. Notably, in comparison to CAR, JUMP directly measures the market's reaction to significant deviations in anticipated earnings, thereby mitigating the influence of return noise. Therefore, we will use the JUMP as the dependent variable, further utilize an ordered probit model to evaluate the efficacy of different earnings surprise indicators, with JUMP as the key explanatory variable.

(1)–(4) in Table 11 are different earnings surprises calculated by the whole sample, which are consistent with the explanatory variables in Table 4, the coefficient of ERROR1 and JUMP is 0.002, and the t-value is 1.03, the coefficient of ERROR2 and JUMP is 0.01, and the t-value is 1.85, which is statistically significant but only at the 10% level; the coefficient of FOM is 0.131, which is statistically significant at 1%, while controlling for the three earnings surprises, FOM remains significant and the other two earnings surprises coefficients are insignificant.

Column(5)–Column (8) compare the earnings surprises calculated based on the mean and latest values of the star analysts' forecasts and the fraction of misses on the same side calculated by all analysts, which is consistent with the method in Table 5, the coefficient of ERROR1, ERROR2 and JUMP are 0.001 and 0.002, respectively, and the t-value is not significant, the coefficient of FOM and JUMP is significant at the level of 1%, the results remain unchanged even though controlling for both ERROR1 and ERROR2. In the final segments, Column (9)–

**Table 10. Earnings surprises: The effect of earnings pre-announcement and preliminary earnings.**

|  | CAR | CAR | CAR | CAR | CAR_NEW | CAR_NEW | CAR_NEW | CAR_NEW |
|---|---|---|---|---|---|---|---|---|
| ERROR1 | 0.016*** |  |  | -0.053** | 0.002 |  |  | 0.002 |
|  | (5.33) |  |  | (-2.34) | (1.48) |  |  | (0.28) |
| ERROR2 |  | 0.074*** |  | 0.193*** |  | 0.006 |  | -0.003 |
|  |  | (2.64) |  | (2.79) |  | (1.35) |  | (-0.17) |
| FOM |  |  | 1.575*** | 1.549*** |  |  | 0.297*** | 0.297*** |
|  |  |  | (14.27) | (13.97) |  |  | (5.42) | (5.43) |
| Asset | 0.228*** | 0.232*** | 0.175** | 0.179** | 0.109*** | 0.109*** | 0.099*** | 0.100*** |
|  | (3.25) | (3.32) | (2.50) | (2.57) | (3.16) | (3.16) | (2.87) | (2.87) |
| BM | -10.535*** | -10.685*** | -9.947*** | -10.144*** | -3.374*** | -3.377*** | -3.270*** | -3.277*** |
|  | (-4.61) | (-4.68) | (-4.46) | (-4.54) | (-3.01) | (-3.00) | (-2.91) | (-2.91) |
| Roa | 2.733* | 2.574 | -1.754 | -1.914 | -0.499 | -0.505 | -1.356 | -1.363 |
|  | (1.71) | (1.61) | (-1.08) | (-1.18) | (-0.58) | (-0.59) | (-1.53) | (-1.53) |
| Lev | 0.137 | 0.140 | -0.016 | -0.000 | -0.305 | -0.304 | -0.334 | -0.335 |
|  | (0.29) | (0.29) | (-0.03) | (-0.00) | (-1.31) | (-1.31) | (-1.43) | (-1.44) |
| Inti | 0.003 | 0.003 | 0.000 | 0.000 | 0.002 | 0.002 | 0.002 | 0.002 |
|  | (0.86) | (0.81) | (0.09) | (0.03) | (1.33) | (1.32) | (1.01) | (1.01) |
| Nature | -0.359** | -0.367** | -0.415*** | -0.428*** | -0.141** | -0.141** | -0.152** | -0.152** |
|  | (-2.21) | (-2.26) | (-2.59) | (-2.68) | (-2.17) | (-2.17) | (-2.33) | (-2.32) |
| Age | -0.004 | -0.004 | -0.004 | -0.004 | 0.004 | 0.004 | 0.004 | 0.004 |
|  | (-0.32) | (-0.32) | (-0.37) | (-0.35) | (0.69) | (0.69) | (0.67) | (0.67) |
| Industry | YES | YES | YES | YES | YES | YES | YES | YES |
| Year | YES | YES | YES | YES | YES | YES | YES | YES |
| Constant | -5.964*** | -6.033*** | -4.082** | -4.166*** | -2.564*** | -2.564*** | -2.214*** | -2.217*** |
|  | (-3.76) | (-3.81) | (-2.57) | (-2.63) | (-3.26) | (-3.26) | (-2.79) | (-2.80) |
| N | 7371 | 7371 | 7371 | 7371 | 7371 | 7371 | 7371 | 7371 |
| R2 | 0.024 | 0.025 | 0.055 | 0.056 | 0.016 | 0.016 | 0.021 | 0.021 |

T values in parentheses,

*, **, *** indicate significant at the 10%, 5%, and 1% levels, respectively.

Column (12) considers the impact of the earnings pre-announcement and the preliminary earnings according to the method in Table 10, the results are also consistent with Table 10, only the coefficient of FOM and JUMP is significantly positive. The above results still show that FOM is a more effective measure of earnings surprises, FOM provides more information on earnings.

The previous section uses the difference between the individual stock return and market return to calculate CAR in the window. Considering the robustness measure of the excess return, we use various methods to calculate the excess return.

1. CAPM adjustment.

$$CAR_{CAPM_{j,t}} = \sum_{k=-1,0,1} \left( R_{j,k} - \beta_j R_{m,k} \right) \tag{12}$$

where $R_{j,k}$ is the actual return of stock j on day k in the year t, $R_{m,k}$ is the weighted market return on day k, and $\beta_j$ is the coefficient of the individual stock and market return on the earnings announcement (-300, -46).

**Table 11. Earnings surprises and stock price jumps.**

| | Full sample | | | | Star analysts | | | | Earnings pre-announcement and preliminary earnings | | | |
|---|---|---|---|---|---|---|---|---|---|---|---|---|
| | (1) | (2) | (3) | (4) | (5) | (6) | (7) | (8) | (9) | (10) | (11) | (12) |
| ERROR1 | 0.002 | | | -0.003 | 0.001 | | | 0.002 | 0.002 | | | -0.000 |
| | (1.03) | | | (-0.79) | (0.34) | | | (0.26) | (0.60) | | | (-0.01) |
| ERROR2 | | 0.010* | | 0.014 | | 0.002 | | -0.003 | | 0.006 | | 0.003 |
| | | (1.85) | | (1.43) | | (0.32) | | (-0.23) | | (0.76) | | (0.14) |
| FOM | | | 0.131*** | 0.129*** | | | 0.110*** | 0.110*** | | | 0.171*** | 0.170*** |
| | | | (6.05) | (5.95) | | | (3.42) | (3.42) | | | (5.32) | (5.27) |
| Asset | 0.057*** | 0.057*** | 0.052*** | 0.052*** | 0.043* | 0.043* | 0.038 | 0.039 | 0.079*** | 0.079*** | 0.073*** | 0.073*** |
| | (3.38) | (3.38) | (3.07) | (3.07) | (1.82) | (1.82) | (1.62) | (1.63) | (3.37) | (3.37) | (3.12) | (3.12) |
| BM | -3.382*** | -3.381*** | -3.301*** | -3.299*** | -3.318*** | -3.315*** | -3.249*** | -3.256*** | -2.449*** | -2.448*** | -2.390*** | -2.393*** |
| | (-5.78) | (-5.77) | (-5.62) | (-5.62) | (-3.95) | (-3.94) | (-3.86) | (-3.86) | (-3.23) | (-3.23) | (-3.15) | (-3.15) |
| Roa | -0.250 | -0.251 | -0.659** | -0.656** | 0.331 | 0.330 | -0.001 | -0.003 | 0.055 | 0.050 | -0.419 | -0.427 |
| | (-0.94) | (-0.94) | (-2.39) | (-2.38) | (0.85) | (0.85) | (-0.00) | (-0.01) | (0.12) | (0.11) | (-0.93) | (-0.94) |
| Lev | -0.215** | -0.215** | -0.238** | -0.238** | -0.162 | -0.162 | -0.192 | -0.192 | -0.204 | -0.203 | -0.218 | -0.218 |
| | (-2.27) | (-2.27) | (-2.51) | (-2.51) | (-1.14) | (-1.14) | (-1.35) | (-1.35) | (-1.40) | (-1.39) | (-1.49) | (-1.49) |
| Inti | 0.003*** | 0.003*** | 0.003*** | 0.003*** | 0.003*** | 0.003*** | 0.003*** | 0.003*** | 0.002* | 0.002* | 0.002 | 0.002 |
| | (5.11) | (5.11) | (4.95) | (4.95) | (3.01) | (3.01) | (2.92) | (2.92) | (1.81) | (1.81) | (1.54) | (1.54) |
| Nature | -0.056 | -0.055 | -0.059* | -0.059* | -0.038 | -0.038 | -0.037 | -0.037 | -0.119** | -0.120** | -0.126*** | -0.126*** |
| | (-1.57) | (-1.57) | (-1.67) | (-1.67) | (-0.74) | (-0.74) | (-0.73) | (-0.73) | (-2.46) | (-2.46) | (-2.58) | (-2.59) |
| Age | 0.002 | 0.002 | 0.002 | 0.002 | 0.005 | 0.005 | 0.005 | 0.005 | 0.005 | 0.005 | 0.005 | 0.005 |
| | (0.66) | (0.66) | (0.65) | (0.66) | (1.44) | (1.44) | (1.45) | (1.45) | (1.37) | (1.37) | (1.34) | (1.33) |
| Industry | YES | YES | YES | YES | YES | YES | YES | YES | YES | YES | YES | YES |
| Year | YES | YES | YES | YES | YES | YES | YES | YES | YES | YES | YES | YES |
| N | 16116 | 16116 | 16116 | 16116 | 7119 | 7119 | 7119 | 7119 | 7371 | 7368 | 7371 | 7368 |
| R2 | 0.032 | 0.033 | 0.036 | 0.036 | 0.031 | 0.031 | 0.033 | 0.033 | 0.029 | 0.029 | 0.034 | 0.034 |

T values in parentheses,

*, **, *** indicate significant at the 10%, 5%, and 1% levels, respectively.

2. Fama-French five-factors adjustment:

$$CAR_{FF5_{j,t}} = \sum_{k=-1,0,1} \left( R_{j,k} - \beta_1 MKT_k - \beta_2 SMB_k - \beta_3 HML_k - \beta_4 RMW_k - \beta_5 CMA_k \right) \quad (13)$$

where MKT, SMB, HML, RMW, and CMA are the factors return corresponding to FF5 factors, respectively, and $\beta_j$ is the coefficient of individual stock on the earnings announcement

3. DGTW adjustment. According to the DGTW method, we adjust the cumulative individual stock return during the earnings announcement by market value, book to market ratio and momentum.

Utilizing the methodology previously delineated, we further examine the three variants of adjusted returns. When JUMP = 0, the return is assigned to 0; when JUMP≠0, the return is assigned to the adjusted return itself, and the above revised market reaction are regressed using each of the three measures of earnings surprises. The results in Table 12 show that no matter what method is used to adjust return in the window, the conclusion is consistent with

**Table 12. Revised market reaction adjustment.**

| | CAR_CAPM_NEW | | | | CAR_FF5_NEW | | | | DGTW_NEW | | | |
|---|---|---|---|---|---|---|---|---|---|---|---|---|
| | (1) | (2) | (3) | (4) | (5) | (6) | (7) | (8) | (9) | (10) | (11) | (12) |
| ERROR1 | 0.003 | | | -0.011 | 0.002 | | | -0.013 | 0.002 | | | -0.011 |
| | (1.47) | | | (-0.96) | (1.48) | | | (-1.14) | (1.48) | | | (-1.09) |
| ERROR2 | | 0.020 | | 0.040 | | 0.021 | | 0.045 | | 0.018 | | 0.039 |
| | | (1.22) | | (1.11) | | (1.23) | | (1.27) | | (1.25) | | (1.23) |
| FOM | | | 0.154*** | 0.150*** | | | 0.140*** | 0.136*** | | | 0.134*** | 0.130*** |
| | | | (4.95) | (4.81) | | | (4.66) | (4.51) | | | (4.47) | (4.34) |
| Asset | 0.091*** | 0.091*** | 0.083*** | 0.083*** | 0.077*** | 0.077*** | 0.070*** | 0.070*** | 0.074*** | 0.074*** | 0.067*** | 0.067*** |
| | (3.95) | (3.96) | (3.60) | (3.60) | (3.46) | (3.47) | (3.14) | (3.13) | (3.30) | (3.31) | (3.00) | (2.99) |
| BM | -3.974*** | -3.975*** | -3.838*** | -3.833*** | -3.560*** | -3.561*** | -3.436*** | -3.430*** | -3.522*** | -3.524*** | -3.404*** | -3.399*** |
| | (-4.96) | (-4.96) | (-4.79) | (-4.79) | (-4.57) | (-4.57) | (-4.41) | (-4.41) | (-4.48) | (-4.48) | (-4.33) | (-4.32) |
| Roa | -0.260** | -0.260** | -0.284** | -0.282** | -0.219* | -0.219* | -0.241* | -0.239* | -0.209 | -0.209 | -0.230* | -0.228* |
| | (-1.97) | (-1.97) | (-2.14) | (-2.13) | (-1.72) | (-1.72) | (-1.89) | (-1.87) | (-1.64) | (-1.64) | (-1.80) | (-1.78) |
| Lev | -0.462 | -0.479 | -0.953** | -0.935** | -0.675* | -0.697* | -1.124*** | -1.100*** | -0.683* | -0.701* | -1.111*** | -1.092*** |
| | (-1.14) | (-1.18) | (-2.27) | (-2.23) | (-1.72) | (-1.77) | (-2.77) | (-2.70) | (-1.72) | (-1.76) | (-2.70) | (-2.65) |
| Inti | 0.051 | 0.049 | 0.013 | 0.011 | 0.080 | 0.077 | 0.045 | 0.043 | 0.087 | 0.085 | 0.054 | 0.052 |
| | (0.19) | (0.18) | (0.05) | (0.04) | (0.28) | (0.27) | (0.15) | (0.15) | (0.31) | (0.30) | (0.19) | (0.18) |
| Nature | 0.028 | 0.028 | 0.022 | 0.022 | 0.025 | 0.025 | 0.019 | 0.019 | 0.035 | 0.035 | 0.029 | 0.029 |
| | (0.65) | (0.65) | (0.50) | (0.50) | (0.61) | (0.61) | (0.46) | (0.46) | (0.83) | (0.83) | (0.69) | (0.69) |
| Age | -0.002 | -0.002 | -0.002 | -0.002 | -0.001 | -0.001 | -0.001 | -0.001 | -0.001 | -0.001 | -0.001 | -0.001 |
| | (-0.44) | (-0.44) | (-0.44) | (-0.42) | (-0.39) | (-0.39) | (-0.40) | (-0.37) | (-0.39) | (-0.40) | (-0.40) | (-0.38) |
| Industry | YES | YES | YES | YES | YES | YES | YES | YES | YES | YES | YES | YES |
| Year | YES | YES | YES | YES | YES | YES | YES | YES | YES | YES | YES | YES |
| Constant | -2.138*** | -2.140*** | -1.895*** | -1.899*** | -1.789*** | -1.792*** | -1.569*** | -1.572*** | -1.713*** | -1.716*** | -1.503*** | -1.505*** |
| | (-3.63) | (-3.63) | (-3.20) | (-3.20) | (-3.13) | (-3.14) | (-2.73) | (-2.74) | (-3.04) | (-3.04) | (-2.65) | (-2.66) |
| N | 15958 | 15958 | 15958 | 15958 | 15958 | 15958 | 15958 | 15958 | 15958 | 15958 | 15958 | 15958 |
| R2 | 0.015 | 0.015 | 0.016 | 0.017 | 0.013 | 0.013 | 0.015 | 0.015 | 0.013 | 0.013 | 0.014 | 0.015 |

the previous paper, that ERROR1 and ERROR2 are unable effectively to proxy for earnings surprises, and FOM can fully explains the earnings information contained in ERROR1 and ERROR2, that FOM is a more effective proxy for earnings surprises.

## 6.3. Earnings surprise standardization: Stock price

Some literatures estimate earnings surprises, often used the closing price of last year to standardize [15], so the new earnings surprises ERROR3 is calculated with the standardized uses the closing price at the end of the last year instead of EPS:

$$ERROR3_{j,t} = \frac{FEPS_{j,t} - MEPS_{j,t}}{Price_{j,t}} \quad (14)$$

Table 13 presents the results from validating ERROR3, employing both CAR and CAR_NEW as dependent variable. The coefficient of ERROR3 and CAR is 10.905 with a t-value of 5.56, and the coefficient of ERROR3 and CAR_NEW is 2.025 with a t-value of 2.84, both significant at the 1% level. A review of Table 4 reveals that when using CAR_NEW as dependent variable, the coefficients for both ERROR1 and ERROR2 are not significant. This suggests that ERROR3 contains more information, it may encompass additional valuation metrics, thereby

**Table 13. Earnings surprises and (revised) market reaction: ERROR3.**

|  | CAR | CAR | CAR | CAR_NEW | CAR_NEW | CAR_NEW |
|---|---|---|---|---|---|---|
| ERROR3 | 10.905*** |  | 1.829 | 2.025*** |  | 0.375 |
|  | (5.56) |  | (1.04) | (2.84) |  | (0.54) |
| FOM |  | 1.052*** | 1.026*** |  | 0.192*** | 0.187*** |
|  |  | (14.37) | (13.34) |  | (5.69) | (5.37) |
| Asset | 0.303*** | 0.261*** | 0.262*** | 0.071*** | 0.063** | 0.063** |
|  | (5.50) | (4.81) | (4.82) | (2.88) | (2.57) | (2.57) |
| BM | -16.458*** | -16.324*** | -16.218*** | -3.897*** | -3.875*** | -3.854*** |
|  | (-8.55) | (-8.66) | (-8.60) | (-4.60) | (-4.59) | (-4.55) |
| Roa | -0.587 | -2.561*** | -2.724*** | -0.855* | -1.210*** | -1.243*** |
|  | (-0.61) | (-2.67) | (-2.80) | (-1.92) | (-2.67) | (-2.72) |
| Lev | -0.418 | -0.697** | -0.677** | -0.172 | -0.223 | -0.219 |
|  | (-1.27) | (-2.13) | (-2.07) | (-1.21) | (-1.56) | (-1.53) |
| Inti | 0.005** | 0.005** | 0.005** | 0.004*** | 0.004*** | 0.004*** |
|  | (2.30) | (2.15) | (2.11) | (3.81) | (3.75) | (3.73) |
| Nature | -0.149 | -0.155 | -0.159 | -0.004 | -0.005 | -0.006 |
|  | (-1.29) | (-1.35) | (-1.38) | (-0.09) | (-0.11) | (-0.12) |
| Age | -0.003 | -0.003 | -0.003 | -0.001 | -0.001 | -0.001 |
|  | (-0.33) | (-0.41) | (-0.41) | (-0.22) | (-0.25) | (-0.25) |
| Industry | YES | YES | YES | YES | YES | YES |
| Year | YES | YES | YES | YES | YES | YES |
| Constant | -7.118*** | -5.867*** | -5.861*** | -1.837*** | -1.610*** | -1.608*** |
|  | (-5.39) | (-4.48) | (-4.48) | (-3.05) | (-2.66) | (-2.66) |
| N | 15860 | 15860 | 15860 | 15860 | 15860 | 15860 |
| R2 | 0.022 | 0.034 | 0.034 | 0.017 | 0.019 | 0.019 |

T values in parentheses,

*, **, *** indicate significant at the 10%, 5%, and 1% levels, respectively.

enhancing the validity of the earnings surprise estimate. However, when FOM is incorporated into the regression analyses, the coefficients associated with ERROR3 are insignificant, indicating that FOM completely absorbs the unexpected earnings information contain in ERROR3. The fraction of misses on the same side as a better proxy for earnings surprises, even though FOM remains the preeminent measure, independent of stock price data. It is debatable whether the calculation of earnings surprises should be standardized by stock price.

## 7. Conclusions

While a substantial body of capital market research relies on measures of earnings surprises, the diversity of estimation methods persists. In this paper, we have conducted an in-depth examination to determine which estimation method serves as a superior proxy for earnings surprises. Recognizing that CAR can be influenced by financial anomalies and may exhibit biases, we introduced a novel approach: constructing the revised market reaction (CAR_NEW) based on stock price jumps. Additionally, we calculated the annual earnings correction (EPS_PUB) by measuring the difference between the earnings reported in the preliminary earnings and those in the annual report, and it acts as the proxy for the earnings surprises. The result shows that CAR cannot validity proxy for the investors' reactions to earnings correction, and suggests that it is reasonable to use the CAR_NEW to proxy the reaction of investors to

earnings. Subsequently, we employ CAR_NEW to evaluate the three measures of earnings surprises and find that only FOM is a valid proxy variable for earnings surprises. We further use high-frequency data to track the trading behavior of investors and find that the investors trade on FOM. We also find that the returns of portfolio constructed by FOM can fully explain that by ERROR.

In summary, our research demonstrates that despite the systematic optimism bias in analysts' earnings forecasts, the nonparametric estimation of analysts' earnings forecasts (FOM) remains a better proxy for earnings surprises. This finding is helpful for future earnings surprises fully explained in related academic fields. We introduce the innovative concept of utilizing stock price jumps to construct a revised market reaction, thereby addressing biases in stock returns associated with financial anomalies during earnings announcements—an aspect often overlooked in existing literature. Furthermore, we use the unique announcements disclosure mechanism in Chinese stock market, and confirm the enhanced precision of the revised market reaction in measuring investor responses, this valuable contribution can assist researchers in recognizing potential market response biases in future investigations. For investors, while absolute values of analysts' earnings forecasts may exhibit significant errors, the fraction of misses on the same side remains a valuable tool, which helps investors make more efficient use of analyst information.

## Supporting information

**S1 Appendix.**
(DOCX)

**S1 Data.**
(RAR)

## Author Contributions

**Conceptualization:** Qin Pan, Kai Huang.

**Data curation:** Qin Pan.

**Formal analysis:** Qin Pan, Kai Huang.

**Funding acquisition:** Kai Huang.

**Investigation:** Qin Pan.

**Methodology:** Qin Pan, Kai Huang.

**Project administration:** Qin Pan.

**Resources:** Qin Pan.

**Validation:** Qin Pan.

**Visualization:** Qin Pan.

**Writing – original draft:** Qin Pan.

**Writing – review & editing:** Kai Huang.

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
