## [Decision Letter · Decision Letter 0]

5 Nov 2023

PONE-D-23-35193How to Measure Earnings Surprises: Based on Revised Market ReactionPLOS ONE

Dear Dr. Pan,

Thank you for submitting your manuscript to PLOS ONE. After careful consideration, we feel that it has merit but does not fully meet PLOS ONE’s publication criteria as it currently stands. Therefore, we invite you to submit a revised version of the manuscript that addresses the points raised during the review process.

We look forward to receiving your revised manuscript.

Kind regards,

Jiafu An

Academic Editor

PLOS ONE

Journal Requirements:

Additional Editor Comments:

You examine the accuracy of earnings surprise measures by considering how the market reacts to earnings announcements. Traditional studies point out that financial market anomalies can affect the calculation of cumulative abnormal returns (CAR) during these announcements. However, the current study presents evidence that a modified market reaction can better capture how investors actually respond to earnings adjustments. You then propose a new method to refine CAR by accounting for sudden stock price movements. I found your research interesting since you introduce the concept of the fraction of misses on the same side (FOM) as a reasonable indicator of earnings surprises. You also discover that investor trading behaviors are consistent with the FOM metric. Notably, investment strategies that exploit the post-earnings announcement drift (PEAD) and utilize FOM are more effective than those based on traditional analysts’ forecast errors.

Since you focus your research on China, you need  to spend some efforts to explain (1) why the Chinese context is particularly suitable to your research question? And (2) is the results generalizable to other empirical settings. To facilitate your motivation, there are several recent studies on China you may want to consider:

An, J., Hou, W., & Zhang, Y. (2019). China’s rule of law in New Era: the rise of regulation and formalism. Journal of Chinese Economic and Business Studies, 17(3), 313-318.An, J., Armitage, S., Hou, W., & Liu, X. (2020). Do checks on bureaucrats improve firm value? Evidence from a natural experiment. Accounting & Finance, 60(5), 4821-4844.

I also strongly recommend the authors to consider adding a “hypothesis development” section to set the theoretical stage of why the authors design such empirical tests. Xu, J. et al. (2023) (Inherited trust and informal finance. Journal of Business Finance & Accounting) offer a good example.

Reviewers' comments:

Reviewer's Responses to Questions

**Comments to the Author**

1. Is the manuscript technically sound, and do the data support the conclusions?

Reviewer #1: Partly

2. Has the statistical analysis been performed appropriately and rigorously? 

Reviewer #1: N/A

3. Have the authors made all data underlying the findings in their manuscript fully available?

Reviewer #1: Yes

4. Is the manuscript presented in an intelligible fashion and written in standard English?

Reviewer #1: Yes

5. Review Comments to the Author

Reviewer #1: Firstly, I would like to ask you to improve the literature review section of your manuscript by incorporating some relevant papers from your publication lists. For example, Yu et al (2022) and Yu and Huang (2023a,b) are relevant to your study as they also examine the impact of external factors on market reactions. By incorporating these papers into your literature review, you can provide a more comprehensive understanding of the external factors that may influence market reactions.

Secondly, we suggest that you provide a more detailed explanation of the methodology used to adjust the cumulative abnormal returns (CAR) using stock price jumps. This will help readers to better understand the approach used in your study. We also recommend that you provide more information on the data used in your study, such as the sample size and the time period covered. This will help readers to better understand the scope of your study. Finally, we suggest that you provide a more detailed discussion of the implications of your findings for investors and analysts.

References:

Yu, D., & Huang, D. (2023a). Cross-sectional uncertainty and expected stock returns. Journal of Empirical Finance, 72, 321–340.

Yu, D., & Huang, D. (2023b). Option-Implied Idiosyncratic Skewness and Expected Returns: Mind the Long Run. Available at SSRN 4323748.

Yu, D., Huang, D., & Chen, L. (2023). Stock return predictability and cyclical movements in valuation ratios. Journal of Empirical Finance, 72, 36–53.

Yu, D., Huang, D., Chen, L., & Li, L. (2023). Forecasting dividend growth: The role of adjusted earnings yield. Economic Modelling, 120, 106188.

6. PLOS authors have the option to publish the peer review history of their article (what does this mean?). If published, this will include your full peer review and any attached files.

Reviewer #1: No

---

## [Author Response · Author response to Decision Letter 0]

8 Dec 2023

Dear Reviewers:

We would like to thank you for your careful reading, helpful comments, and constructive suggestions, which has significantly improved the presentation of our manuscript. We have carefully considered all comments from the reviewers and revised our manuscript accordingly. In the following section, we summarize our responses to each comment from the reviewers. We believe that our responses have well addressed all concerns from the reviewers. We hope our revised manuscript can be accepted for publication.

Responds to the editor’s comments: 

Response to comment: why the Chinese context is particularly suitable to your research question? 

Response: Thank you for your suggestions. We think the short selling and the unique announcement disclosure mechanism in the Chinese A-share market are suitable to my research.

First, short selling in the Chinese A-share market is substantially limited, for example, short selling was banned prior to 2010 and 2016 restrictions prohibited the short selling of nearly 70% of stocks (Titman et al., 2022). These short-selling restrictions result that the market lacks a short selling mechanism and cannot obtain returns through short selling. Then analysts are required to furnish a higher number of "buy" rating reports to enhance commission income, thereby leading to a stronger optimistic bias (Jackson, 2005). 

Second, due to the unique announcement disclosure mechanism in China, some companies are compelled to disclose the preliminary earnings by CSRC. About half of the listed companies opt to disclose the preliminary earnings before the end of February, then release the annual report around April. Although the preliminary earnings are not audited, they are usually seemed as an important reference to the actual earnings by investors with high accuracy (Liu, 2005). Therefore, we can calculate the difference between the preliminary earnings and the actual earnings as the earnings correction (EPS_PUB), which be the proxy for earnings surprises for specified firms.

Response to comment: the results generalizable to other empirical settings.

Response: Thank you for your suggestions. The results generalizable to other empirical settings as follows.

First, CAR are widely used in the finance research on stock markets (Brown et al., 2009), but ignore the effect of financial anomalies on individual stock returns around the earnings announcement (Engelberg et al., 2018). As far as we know, this is the first paper to point out such problem and prove that the market reaction measured by cumulative abnormal return cannot effectively represent the investors' response to earnings correction. Then we build an appropriate metric for the actual investors’ response to earnings shocks, i.e. revised market reaction (CAR_NEW), and verify that the revised market reaction can be a better measure of the investors' response to earnings correction.

Second, earnings surprises are widely used in the basic research on stock markets such as market efficiency and earning management. For example, earnings surprises can be used to explain large fluctuations in stock prices after earnings announcements, are also commonly used to study the effectiveness of stock market reactions to earnings information; Further, the extent to which investors react to earnings surprises can also measure the level of the earnings quality (Foster et al., 1984; Brown et al., 2009). We prove that FOM is a better proxy for earnings surprises in China’s stock market rather than the analysts’ forecast error to measure earnings surprises as the market consensus in literature uses (Mergenthaler et al., 2011; Bouchaud et al., 2019).

Therefore, future research could replace CAR with CAR_NEW, and proxy earnings surprises with FOM. These conclusions have special value to academic research. Some studies using CAR proxy for investor reactions, that may contain noise, which may lead to different results. The average or the latest analysts’ forecast cannot proxy for market consensus, which are widely used in the basic research, may also bring different conclusions.

Response to comment: to facilitate your motivation, there are several recent studies on China you may want to consider:

An, J., Hou, W., & Zhang, Y. (2019). China’s rule of law in New Era: the rise of regulation and formalism. Journal of Chinese Economic and Business Studies, 17(3), 313-318.

An, J., Armitage, S., Hou, W., & Liu, X. (2020). Do checks on bureaucrats improve firm value? Evidence from a natural experiment. Accounting & Finance, 60(5), 4821-4844.

Response: Thank you for your suggestions. We incorporated these papers into our introduction, please see the newly revised manuscript for more details (1. Introduction and 4.4. Other Control Variables).

Response to comment: I also strongly recommend the authors to consider adding a “hypothesis development” section to set the theoretical stage of why the authors design such empirical tests. Xu, J. et al. (2023) (Inherited trust and informal finance. Journal of Business Finance & Accounting) offer a good example.

Response: Thank you for your suggestions. We realize that it is necessary to adding a “hypothesis development”, so we have added these parts. Please see the newly revised manuscript for more details (2. Hypothesis Development). The following is a brief overview.

First, due to CAR needs to be amended to eliminate the effects of financial anomalies, we use stock price jumps as a proxy for information shocks and construct corrective market reactions (CAR_NEW). Then we examine whether CAR or CAR_NEW is a better proxy for investors' reactions to earnings information.

Second, we explore the correlation between various measures of earnings surprises and CAR_NEW to test the accuracy of various earnings surprise measures.

Third, we examine the validity of earnings surprises from other two aspects. On the one hand, we use high-frequency data of investors' trading to capture investors' active behavior, examine whether investor trading behavior is significantly related to earnings surprises around earnings announcement. On the other hand, we employ a portfolio-based approach to test whether different measures of earnings surprises obtain difference PEAD.

Responds to the reviewer’s comments: 

Response to comment: Firstly, I would like to ask you to improve the literature review section of your manuscript by incorporating some relevant papers from your publication lists. For example, Yu et al (2022) and Yu and Huang (2023a,b) are relevant to your study as they also examine the impact of external factors on market reactions. By incorporating these papers into your literature review, you can provide a more comprehensive understanding of the external factors that may influence market reactions.

Response: Thank you for your suggestions. We incorporated these papers into our introduction, please see the newly revised manuscript for more details (1. Introduction and 4.4. Other Control Variables).

Response to comment: a more detailed explanation of the methodology used to adjust the cumulative abnormal returns (CAR) using stock price jumps

Response: Thank you for your suggestions. We show the methodology used to adjust the cumulative abnormal returns (CAR) using stock price jumps as follow.

Some studies have revealed the substantial influence of financial anomalies on individual stock returns around earnings announcement (Saver and Wilson (2016); Engelberg et al. (2018)). Then, we find financial anomalies potentially introducing bias into market reaction measurements, and we believe that CAR needs to be amended to eliminate the effects of financial anomalies.

According to the Efficient Market Hypothesis, stock prices follow a random walk and exhibit price jumps in response to earnings shocks. Jiang and Zhu (2017) demonstrated that stock prices experience jumps in response to significant information shocks, therefore, we use stock price jumps as a proxy for information shocks. The test statistics of stock price jumps are defined by Jiang and Oomen (2008) and Jiang and Yao (2013):

(V_((0,T) ) N)/√(Ω_SWV )(1-(RV_N)/(SWV_N ) )^d→N(0,1)

If the stock price exhibits a positive jump during the annual earnings announcement (- 1,1), then JUMP=1; If there is a negative jump in the stock price during the annual earnings announcement (- 1,1), then JUMP=-1; otherwise, JUMP=0.

We consider that CAR cannot effectively represent the reaction of investors' response to earnings information, use stock price jumps to proxy the information shocks, and introduce an innovative revision to market reaction, denoted as CAR_ NEW incorporating stock price jumps. Revised market reaction(CAR_NEW) as follow:

CAR_NEW={█(CAR ,Jump≠0@0 ,Jump=0)┤

while the stock price jumps, there is a significant earnings surprises around the earnings announcement, and the revised market reaction is the market reaction itself, while the stock price no jump, the return during the announcement date due to earnings information obeys a random wander and the revised market reaction takes the value of zero.

Response to comment: the data used in your study.

Response: Thank you for your suggestions. We obtain daily stock return, analysts' earnings forecasts, earning announcement data and basic financial data for China A-shares from the China Stock Market and Accounting Research (CSMAR) database (https://data.csmar.com/). CSMAR is a comprehensive research-oriented database focusing on China Finance and Economy and based on academic research needs, meeting with the international professional standards while adapting to China’s features. Our sample starts in January of 2001; the Chinese stock market is relatively undeveloped before then, and has more accuracy data of analysts' data. Our sample of earning announcements ends in June 2021, the analysts' earnings forecasts ends in April 2021 and the daily stock return ends in June 2021. 

As analysts predominantly forecast annual earnings for stocks, so our analysis is limited to annual earnings data. We process the sample data as follows: (1) retain only the annual analysts' earnings forecasts made within 1 years of the annual announcement, (2) exclude samples with less than 5 analysts' earnings forecasts for each year; (3) exclude samples with less than 1 year of presence on the A -share; (4) exclude companies from the financial industry; (5) exclude samples with missing variable data. To mitigate the potential influence of outliers on our conclusions, we winsorized all core continuous variables at the 1st and 99th percentiles. Ultimately, our sample comprises 16,116 company-year observations. Table 1 shows the descriptive statistics results of the core variables.

Table 1. Descriptive statistics

Variables N MEAN STD P1 P25 MED P75 P99

EPS_PUB 7142 -0.016 0.375 -0.333 -0.01 0 0 0.145

FOM 16116 -0.394 0.651 -1 -1 -0.657 0.071 1

ERROR1 16116 -0.536 5.278 -9.077 -0.41 -0.122 0.007 1.597

ERROR2 16116 -0.155 2.061 -3.353 -0.109 -0.004 0.027 0.940

ERROR3 15860 -0.01 0.027 -0.12 -0.012 -0.004 0 0.025

JUMP 16116 -0.021 0.33 -1 0 0 0 1

CAR 16116 0.044 5.391 -12.480 -3.020 -0.400 2.640 16.810

CAR_NEW 16116 -0.048 2.437 -9.060 0.000 0.000 0.000 9.470

Asset 16116 22.592 1.364 20.231 21.604 22.381 23.372 26.688

BM 16116 0.042 0.033 0.006 0.021 0.033 0.053 0.186

Roa 16116 0.086 0.058 -0.058 0.047 0.076 0.113 0.292

Lev 16116 0.434 0.197 0.058 0.277 0.433 0.587 0.847

Inti 16116 50.962 24.814 1.522 32.25 55.373 70.59 93.206

Nature 16116 0.404 0.491 0 0 0 1 1

Age 16116 2.058 0.776 0 1.386 2.197 2.708 3.258

Response to comment: provide a more detailed discussion of the implications of your findings for investors and analysts.

Response: Thank you for your suggestions. The systematic optimistic bias leads to analysts' overestimation of earnings, and earnings surprises estimated based on analysts' earnings forecasts are higher than market consensus. We prove that using FOM (the mean of analysts' earnings forecasts' signs) as a proxy for earnings surprises can solve the systematic analyst bias, and that investor behavior is also consistent with FOM. 

We examine the correlation between earnings surprises and investor trading behavior. First, the high-frequency data to track the trading behavior of investors shows that only FOM exhibits a substantial positive correlation with order imbalance, i.e., investors show a preference for buying over selling when FOM is positive, which provides new evidence that FOM is a robust measure of earnings surprises. Second, we investigate PEAD strategies for different earnings surprise and find that the premium of PEAD constructed by FOM can fully explain that by ERROR (the average or the latest analysts’ forecast). These results also show that FOM is a more validity measure of earnings surprises.

Therefore, investor should pay attention on the consistency of analyst forecast propensity, which have more information of earnings forecast.

References

 Jackson A R. Trade generation, reputation, and sell‐side analysts[J]. The journal of finance, 2005, 60(2): 673-717. 

 Titman S, Wei C, Zhao B. Corporate actions and the manipulation of retail investors in China: An analysis of stock splits[J]. Journal of Financial Economics, 2022, 145(3): 762-787.

 Jackson A R. Trade generation, reputation, and sell‐side analysts[J]. The journal of finance, 2005, 60(2): 673-717.

 Muhua L .The Information Content of Preliminary Earnings Estimate: Empirical Evidences and Policy Suggestions[J].accounting research, 2005.

 Brown S, Hillegeist S A, Lo K. The effect of earnings surprises on information asymmetry[J]. Journal of Accounting and Economics, 2009, 47(3): 208-225.

 Engelberg J, McLean R D, Pontiff J. Anomalies and news[J]. The Journal of Finance, 2018, 73(5): 1971-2001.

 Foster G, Olsen C, Shevlin T. Earnings releases, anomalies, and the behavior of security returns[J]. Accounting Review, 1984: 574-603.

 Mergenthaler R, Rajgopal S, Srinivasan S. CEO and CFO career penalties to missing quarterly analysts forecasts[J]. Available at SSRN 1152421, 2012.

 Bouchaud J P, Krueger P, Landier A, et al. Sticky expectations and the profitability anomaly[J]. The Journal of Finance, 2019, 74(2): 639-674.

 Savor P, Wilson M. Earnings announcements and systematic risk[J]. The Journal of Finance, 2016, 71(1): 83-138.

 Jiang G J, Zhu K X. Information shocks and short-term market underreaction[J]. Journal of Financial Economics, 2017, 124(1): 43-64.

 Jiang G J, Oomen R C A. Testing for jumps when asset prices are observed with noise–a “swap variance” approach[J]. Journal of Econometrics, 2008, 144(2): 352-370.

 Jiang G J, Yao T. Stock price jumps and cross-sectional return predictability[J]. Journal of Financial and Quantitative Analysis, 2013, 48(5): 1519-1544.

---

## [Editor Report · Decision Letter 1]

13 Dec 2023

How to Measure Earnings Surprises: Based on Revised Market Reaction

PONE-D-23-35193R1

Dear Dr. Pan,

We’re pleased to inform you that your manuscript has been judged scientifically suitable for publication and will be formally accepted for publication once it meets all outstanding technical requirements.

Kind regards,

Jiafu An

Academic Editor

PLOS ONE

Additional Editor Comments (optional):

I think the authors have adequately addressed the reviewer and my comments. Therefore I am happy to accept the paper.
---

## [Editor Report · Acceptance letter]

14 Dec 2023

PONE-D-23-35193R1 

PLOS ONE

Dear Dr. Pan, 

I'm pleased to inform you that your manuscript has been deemed suitable for publication in PLOS ONE. Congratulations! Your manuscript is now being handed over to our production team.

Kind regards, 

on behalf of

Dr. Jiafu An 

Academic Editor

PLOS ONE